# Maternal Diet Quality and the Health Status of Newborns

**DOI:** 10.3390/foods11233893

**Published:** 2022-12-02

**Authors:** Jitka Pavlikova, Antonin Ambroz, Katerina Honkova, Irena Chvojkova, Radim J. Sram, Pavel Rossner, Jan Topinka, Tomas Gramblicka, Ondrej Parizek, Denisa Parizkova, Jana Schmuczerova, Jana Pulkrabova, Andrea Rossnerova

**Affiliations:** 1Institute of Experimental Medicine AS CR, Department of Genetic Toxicology and Epigenetics, Videnska 1083, 142 20 Prague, Czech Republic; 2Institute of Experimental Medicine AS CR, Department of Nanotoxicology and Molecular Epidemiology, Videnska 1083, 142 20 Prague, Czech Republic; 3Department of Food Analysis and Nutrition, Faculty of Food and Biochemical Technology, University of Chemistry and Technology, Technicka 3, 166 28 Prague, Czech Republic; 4Department of Medical Genetics, L. Pasteur University Hospital, Trieda SNP 1, 040 11 Kosice, Slovakia

**Keywords:** 8-isoprostane, birth weight, DDT, maternal diet quality, maternal protein intake, oxidative stress, persistent organic pollutants

## Abstract

The maternal diet during pregnancy affects neonatal health status. The objective of this study was to assess the nutritional quality of the maternal diet, and its contamination by persistent organic pollutants (POPs), in pregnant women living in two areas of the Czech Republic with different levels of air pollution, and subsequently to assess the relationship of these two factors with birth weight and neonatal oxidative stress. To determine the level of oxidative stress, 8-isoprostane concentrations in umbilical cord plasma were measured. The overall nutritional quality of the maternal diet was not optimal. Of the nutritional factors, protein intake proved to be the most significant showing a positive relationship with birth weight, and a negative relationship with the oxidative stress of newborns. Dietary contamination by persistent organic pollutants was low and showed no statistically significant relationship with birth weight. Only one of the 67 analyzed POPs, namely the insecticide dichlorodiphenyltrichloroethane (DDT), showed a statistically significant positive relationship with the level of neonatal oxidative stress.

## 1. Introduction

It has been shown that lower birth weight is connected with chronic diseases in later life. Inadequate nutrition during pregnancy results in fetal adaptation to the environment with a limited supply of nutrients. A metabolism with these settings is later unable to process the glut of easily accessible food [1]. This thrifty phenotype hypothesis was confirmed in 1997 by a study with monozygotic and dizygotic twins, which showed the dependence of type 2 diabetes on birth weight rather than on the genotype [2]. Subsequent epidemiological studies began to address the relationship between health status and maternal diet during pregnancy. It has been hypothesized that environmental conditions during pregnancy permanently affect the epigenome and, consequently, health later in life [3]. These effects may be passed down to future generations [4,5,6].

The intrauterine development is affected not only by the nutritional composition of the maternal diet, but also by the presence of toxic substances in food, such as, e.g., benzo[a]pyrene or acrylamide. Maternal nutrition significantly affects the formation and function of the placenta [7,8,9] including the expression of CYP1A1, an enzyme involved in xenobiotic and drug metabolism [10]. The maternal diet also impacts the gut microbiota, the composition of which has a significant effect on the metabolism of xenobiotics and vice versa [11,12].

Maternal diet may further affect oxidative stress in newborns [13] resulting in damage to macromolecules in the human body, e.g., lipids in cell membranes or DNA [14]. High levels of oxidative stress can contribute to the development of many serious neonatal diseases [15].

In our research, we have focused on the relationship between the maternal diet during pregnancy and the birth weight and neonatal levels of 8-isoprostane, an established biomarker of oxidative stress. Birth weight appears to be related to health in later life. We evaluated the maternal diet quality both in terms of its nutritional composition, and its contamination with persistent organic pollutants (POPs). We analyzed the following: energy intake, intake of individual macronutrients and nutrient groups, dietary diversity, distribution of food intake during the day, preference of low-fat food variants, and intake of contaminants from five different groups of persistent organic pollutants (polychlorinated biphenyls, organochlorine pesticides, brominated flame retardants, perfluorinated compounds, and polycyclic aromatic hydrocarbons). We hypothesized that birth weight would be associated primarily with energy and protein intake, while oxidative damage would be associated with fat, fruit, vegetable, and POP intake.

## 2. Materials and Methods

### 2.1. Participants

The participants of the study were 54 mothers and their newborns living in two regions of the Czech Republic. The industrial region of Karvina is located in northern Moravia (48% participants), and the agricultural region of Ceske Budejovice lies in Southern Bohemia (52% participants). The mothers were recruited from an antenatal course for pregnant women in Karvina, and during prenatal visits at the Hospital Ceske Budejovice. Participation in the research was voluntary. Mothers signed an informed consent form approved by the Ethics Committees of the University of Ostrava (OU-87633/90-2018, 30 November 2018), Hospital of Ceske Budejovice, a.s. (101/19, 4 January 2019), Hospital Karvina-Raj (16,827, 25 October 2018). The study was conducted during the years 2019 and 2020. Only mothers who reported a non-smoking status were included in the study.

### 2.2. Study Design

Pregnant women recorded food and drink consumption on pre-prepared forms for seven consecutive days, in the period of two weeks prior to their birth due date. One-fourth of each serving was stored in a plastic box. The servings from each day were combined in a single box and stored in a freezer. The boxes were later transported to the University of Chemistry and Technology, Prague, where the content of each box was homogenized and further treated as one food sample. In these samples, the concentration of 67 different substances belonging to five groups of persistent organic pollutants (POPs) was measured: polychlorinated biphenyls (PCBs-8 substances), organochlorine pesticides (OCPs-10 substances), brominated flame retardants (BFRs-11 substances), perfluorinated alkylated substances (PFAS-15 substances), and polycyclic aromatic hydrocarbons (PAHs-23 substances) (Appendix A). 

Socioeconomic data and data on maternal height, pre-pregnancy weight, weight before delivery, and pre-pregnancy waist circumference were obtained in personal interviews with the mothers during their stay in the maternity hospitals. Medical data on the course of pregnancy, childbirth, and the health status of newborns were obtained with the consent of mothers from their health cards in the maternity hospitals.

### 2.3. Dietary Assessment

The database “calorie tables” (kaloricketabulky.cz) and the questionnaire “Control practical test of nutritional quality of pregnant and lactating women” (CPT) were used to acquire the data on maternal diet quality [16,17]. Therefore, 16 indicators characterizing the diet of each day were obtained. The database “kaloricketabulky.cz” is a Czech food database containing information on food composition and energy value. This database was used to obtain data on energy, protein, carbohydrate, fat, and fiber intake. If a particular food was not in the database, it was replaced by the most similar equivalent.

The questionnaire CPT evaluates the diet quality over a 24 h period and consists of ten ‘yes/no questions’. These questions relate to food intake from different food groups, time distribution of food intake, diet diversity, and low-fat food preferences for each day (Table 1 shows the cut points for each queried qualitative factor). The total positive answers (= point gain) then indicate the overall diet quality for a given day.

For each woman, the average values of diet factors obtained from the database “kaloricketabulky.cz” from all of her monitored days were calculated. The values of diet factors obtained from CPT were calculated as the percentage of days in which the woman reached or exceeded the cut point. Next, the overall diet quality was calculated for each woman as her average point gain in CPT from all the days when her diet quality was monitored.

### 2.4. Persistent Organic Pollutant Analysis

A method for the determination of PAHs and selected persistent organic pollutants (PCBs, OCPs, and PBDEs) in the diet is described elsewhere [18]. Briefly, the GC-amenable contaminants were extracted into ethyl-acetate with the support of inorganic salts. Hand-made silica columns were employed for the clean-up stage. For the separation and identification of PCBs and OCPs gas chromatography, coupled to tandem mass spectrometry operated in electron impact ionization (GC-EI-MS/MS) on a capillary column DB-5MS (30 m × 0.25 mm × 0.25 µm), was utilized. PAHs were analyzed using GC-EI-MS/MS on a capillary column Rxi-PAH (40 m × 0.18 mm × 0.07 µm). Selected PBDEs were analyzed using GC coupled to mass spectrometry in negative chemical ionization (GC-NCI-MS), with separation on a capillary column DB-XLB (15 m × 0.25 mm × 0.10 µm). The LOQs for the abovementioned compounds ranged between 0.003 and 0.05 ng/g ww.

Selected PFAS, PBP, and isomers of HBCDD were isolated by acetonitrile supported with inorganic salts. Dispersive solid phase extraction (dSPE) using sorbent zSEP+ was used for the clean-up stage. Target PFAS were analyzed using ultra-high performance liquid chromatography coupled to tandem mass spectrometry, operated in electrospray ionization in negative mode (UHPLC-ESI-MS/MS). The target compounds were separated on a reverse phase employing a BEH C18 column (100 × 2.1 mm; 1.7 µm). The LOQs of PFAS, PBP, and HBCDDs were in the range of 0.01–0.1 ng/g ww.

### 2.5. Oxidative Stress Analysis in Mothers and Newborns

To determine the level of oxidative stress, 8-isoprostane concentrations were measured in maternal and umbilical cord plasma samples. Umbilical cord plasma samples were obtained immediately after birth. Maternal plasma samples were collected during the mother’s stay in the maternity hospital in the Department of Obstetrics. Frozen samples were transferred to the Institute of Experimental Medicine CAS, where 8-isoprostanes levels were measured using immunoassay kits from Cayman Chemical Company (Ann Arbor, MI, USA) according to the manufacturer’s protocol [19]. In total, 125 µL of plasma samples were used for these analyses.

### 2.6. Statistical Analysis

Statistical analysis was performed using software R (version 3.6.1). The normality of distribution was evaluated by the Shapiro–Wilk test. Data not meeting the normal distribution were converted to log-values. If normal distribution was still not achieved, the relevant hypotheses were tested by non-parametric tests. However, in the case of a large data set (at least one hundred observations), parametric tests were used, even in the case of non-normal distributions. Multiple regression models were used to perform the statistical analysis of the associations of dietary factors with birth weight and neonatal 8-isoprostane levels.

Statistical analysis was performed for both the individual persistent organic pollutants and their groups. Ten POP groups were created for use in the statistical analysis (Appendix A). 

## 3. Results

### 3.1. Participants

The basic characteristics of the studied groups (mothers and their newborns) are summarized in Table 2.

All the mothers were of European origin. The mean age of all mothers from both regions was 31 years (range = 22–43). All the mothers reported no significant financial problems, and 61% of them were university graduates. All the mothers self-reported as non-smokers, and four mothers (three mothers from Ceske Budejovice and one mother from Karvina) reported exposure to second-hand smoke at the time of sampling. Cotinine levels were under the limit of quantification (LOQ) in 40 mothers (75%); the maximum level was 39 ng/mg creatinine. Fifty-four percent of mothers reported vitamin and/or mineral supplement intake during the last trimester of pregnancy. The mean pre-pregnancy weight was 68 kg (median 66 kg, range 43–115 kg), the mean pre-pregnancy BMI was 24 kg/m^2^ (median 23 kg/m^2^, range 17–41 kg/m^2^), and the mean pre-pregnancy waist-to-height ratio (WHtR) was 0.47 (median 0.45, range 0.36–0.80). According to this parameter, pre-pregnancy weight was normal in 65% of mothers (*n* = 35), and 26% of mothers (*n* = 14) were overweight and 9% of mothers (*n* = 5) were obese.

All the pregnancies were singleton, 43% of mothers were primiparous and no pregnancy was terminated by preterm birth. The majority of births were vaginal deliveries (72%).

All the children were born between the 37th and 41st week of pregnancy with normal birth weight, (mean 3440 g (range 2650–4580 g)). Forty-five newborns (83%) had an Apgar score of 10, with seven newborns 9 and two newborns 8. (Table 2) Fifty-four percent of infants were males.

### 3.2. Diet Quality

Three hundred thirty-one daily food records (178 in Ceske Budejovice, 153 in Karvina) from fifty-one mothers were collected and analyzed in this study.

The results of the analysis of energy and macronutrient intake are summarized in Figure 1. The average daily protein intake did not reach the recommended value of 80 g [20] in 63% of the mothers. For fiber, the recommended value of 25 g [21] was not reached by 90% of mothers.

Figure 2 shows the food intake from individual nutritional groups, diet diversity, time distribution of food intake, and preferences for low-fat food. The average daily intake of raw vegetables and dairy products was insufficient for all mothers.

The average maternal overall diet quality ranged from 2.3 to 9 points out of a maximum of 10 points.

Protein, carbohydrate, fiber, and cereal intake were significantly higher in mothers with a university degree. These mothers also had better time distribution of food intake, greater preference for low-fat foods, and better overall diet quality [Table 3].

Maternal age, pre-pregnancy weight, and pre-pregnancy WHtR did not show a significant association with any dietary factor.

### 3.3. Diet POPs

Three hundred fifty-two food samples (178 in Ceske Budejovice, 174 in Karvina) were collected from fifty-four mothers. The daily POPs intake originating from a solid food was calculated by multiplying the POPs concentration in the food sample and the total weight of the consumed solid food per day. The highest daily ∑PCBs, ∑PBDE congeners, ∑HBCD isomers, and ∑PAHs intakes ranged in tens of µg, ∑DDT isomers, and their metabolites intakes ranged in units of µg, HCB, ∑HCH isomers, ∑PBDE congeners, PBP, ∑perfluorinated sulfonates, and ∑perfluorinated carboxylic acids intake ranged in tenths of µg. Figure 3 shows the daily intake of all POPs groups.

From the group of seven PCBs (six non-dioxin-like + PCB 118), the congeners with higher molecular weight (PCB 118, 138, 153, and 180) were found in a significant number of samples, i.e., in 51%, 90%, 93%, and 71% of samples, respectively. Their concentrations (in ng/g wet weight, min-max/median) were as follows: PCB 118 (<0.003 to 0.435/0.003), PCB 138 (<0.003 to 4.27/0.014), PCB 153 (<0.003 to 5.56/0.018), and PCB 180 (<0.003 to 2.92/0.009). The remaining PCBs were detected at very low concentrations in less than 30% of samples. The sum of six NDL-PCBs ranged between <0.005 and 13.0 ng/g ww with a median of 0.044 ng/g ww.

Selected OCPs were found in the same range of concentrations compared to PCBs. HCB, p,p′-DDT, and its metabolites (p,p′-DDE and p,p′-DDD) were found in more than 50% of samples, specifically 98%, 53%, 99%, and 79%, respectively. The highest amounts were observed for p,p′-DDE (<0.003 to 3.96/0.133) followed by HCB (0.003–0.609/0.038), p,p′-DDD (<0.003 to 0.347/0.007), and p,p′-DDT (<0.005 to 1.03/0.006). The total of DDTs was <0.005 to 5.43 with a median of 0.160 ng/g ww.

Slightly lower concentrations compared to the two aforementioned groups of POPs were observed for the analyzed BFRs, where only PBDE congeners 47 and 99 were found in a significant portion of samples, i.e., 64% and 76%. BDE 47 was found in the amounts <0.003 to 0.762/0.005 and BDE 99 <0.003 to 1.76/0.012. The total of all analyzed PBDEs (*n* = 20) was in the range <0.05 to 50.3 ng/g ww. It should be noted that the outlier maximum value belonged to PBDE 209, which was the only congener found in this sample, and the total detection frequency of PBDE 209 was 7%.

The concentration of PFAS as well as their detection frequencies were generally very low. The two predominant compounds from this group were PFOA (<0.006 to 0.117 ng/g ww) and PFOS (<0.006 to 0.301), which were detected in 16% and 9%, respectively. The remaining PFAS were detected in only 1–5% of samples, or they were not detected at all.

The last group of analyzed pollutants was PAHs, specifically the lighter ones, e.g., ACE, PHE, FLT, and PYR, which belonged to the highest measured. All of the 3–4 ring PAHs were detected in more than 60% of samples and their concentrations were one order of magnitude higher than those of PCBs and OCPs, e.g., PHE (<0.05 to 14.3/0.639), PYR (<0.05 to 25.8/0.178), FLT (<0.05 to 11.4/0.153), and ACE (<0.05 to 3.58/0.081). These PAHs comprised relatively more than 70% of all PAHs. The carcinogenic BaP was detected in only 12% of samples at concentrations <0.05 to 4.56 ng/g ww and a total of four carcinogenic PAHs (BaA, BaP, BbFA, and CHR) was positively detected in 19% of samples at concentrations <0.05 to 4.56 ng/g ww.

### 3.4. Birth Weight

Energy intake, protein, and meat intake showed a positive association with birth weight in our cohort, but the relationship was only statistically significant for protein intake (*p* = 0.011). Using the multiple regression model, it was found that 25% (22% Adjusted R-squared) of the birth weight value variation was explained jointly by maternal pre-pregnancy weight and protein intake (*p* = 0.001). Both factors showed a positive association with birth weight, and contribute similarly to the regression model (protein intake *p* = 0.009, pre-pregnancy weight *p* = 0.007).

Birth weight had no relationship with any POP food intake in our cohort. Birth weight also had no relationship with maternal weight gain.

### 3.5. Plasma 8-Isoprostane

8-isoprostane concentrations were determined in both neonatal and maternal plasma samples. Their levels did not significantly differ between mothers and their newborns in pair tests. Maternal 8-isoprostane levels were in the range of 18–413 pg/mL of plasma with a mean of 120 pg/mL, neonatal 8-isoprostane levels were in the range of 31–385 pg/mL of plasma with a mean of 115 pg/mL. Neither the maternal nor neonatal levels differed between localities [Figure 4].

Using the multiple regression model, it was found that 42% (40% Adjusted R -squared) of neonatal 8-isoprostane levels variability was explained by pp-DDT and protein intake (*p* < 0.001). The pp-DDT intake had a much stronger effect in this regression model (*p* = 0.0000047), but the protein intake contribution was also statistically significant (*p* = 0.00044) [Figure 5].

No significant relationship was found between neonatal 8-isoprostane levels and fruit or vegetable intake (Figure 6).

## 4. Discussion

There were two main aims of our study. First, to examine the diet quality of pregnant women, including exposure to POPs, in the Czech Republic. Second, to analyze the relationship between maternal diet during pregnancy (including POPs intake) and the birth weight and neonatal levels of oxidative stress.

The results of our study indicate that the diet quality of Czech pregnant women does not meet the recommended standards. In total, 63% of mothers did not have sufficient protein intake and 90% of mothers had low fiber intake. In our cohort, there were no mothers with an adequate intake of raw vegetables and dairy products. In the Czech Republic, good quality food is available all year round in a very dense retail network. The Czech Republic is a developed welfare state with many social benefits. The vast majority of the population has the financial means to buy good quality food. All study participants were asked about their financial situation. No one reported financial problems that would force them to limit their living needs. Thus, there are no external obstacles preventing women from accessing good quality food here. However, it should be taken into account that we only analyzed the diet of a small number of subjects over a short period of time. The question is whether the results are representative of the entire population.

According to the literature, the most significant maternal dietary factors influencing birth weight include energy intake, protein, and meat intake [22,23,24,25]. These factors (meat intake was included in one common factor with egg and legume intake) also showed a positive association with birth weight in our cohort, but the relationship was only statistically significant for protein intake. Other dietary characteristics did not show a significant relationship with birth weight values. Pre-pregnancy BMI is also known to be a significant predictor of birth weight [26,27]. However, the maternal pre-pregnancy weight effect was much stronger than the maternal pre-pregnancy BMI effect.

It can be assumed that the neonatal levels of oxidative stress are affected, among other factors, by the consumption of fruit, vegetables (especially in the raw state), and fats [28,29,30,31,32]. However, no specific food group showed any significant relationship with the maternal or neonatal 8-isoprostane levels in our cohort. The only statistically significant relationship was observed for protein intake: higher than average maternal protein intake was associated with lower values of neonatal 8-isoprostane.

As far as persistent organic pollutant food intake is concerned, the POP content in the analyzed samples was within the recommended limits. The only exception was observed for PFAS. The tolerable weekly intake (TWI) of 4.4 ng/kg body weight per week was determined for ∑PFOS + PFOA + PFNA + PFHxS. Dietary exposure in many European countries exceeds this limit [33,34]. Of the 44 mothers from whom the food samples of all seven days were obtained, the PFAS TWI was exceeded in three mothers (two mothers from Ceske Budejovice and one mother from Karvina). PFAS TWI was also exceeded in one mother from Ceske Budejovice who collected food samples for only 4 days. During these 4 days, her ∑PFOS + PFOA + PFNA + PFHxS dietary intake was 5.26 ng/kg body weight. PFAS are currently used in many industries to repel water and grease from paper food packaging, outdoor clothing, and also carpets and fire-fighting foams, for example. They are known primarily for use in Teflon and Gore-Tex. PFAS are not metabolized, and they accumulate in tissue. Humans obtain PFAS mainly in food and water.

The average daily intake of the sum of PCBs was 0.003 µg/kg body weight (max 0.47 µg/kg b.w.). PCBs are very stable hydrophobic compounds that are poorly soluble in water. They are not subject to biodegradation, so they accumulate in the environment and in organisms. The reason for the low PCBs food intake in our cohort is most likely the fact that the production of PCBs was banned in 2004 by the Stockholm Convention, which has been ratified by 184 countries around the world [35]. In the Czech Republic, the production of these harmful substances was discontinued in 1984. It is due to these interventions that PCB concentrations in the environment, and thus in food, are gradually declining.

The average daily food intake of the total DDT and its metabolites was 0.004 µg/kg body weight (max 0.058 µg/kg body weight), which was orders of magnitude lower than the provisional tolerable daily intake (PTDI) 0.01 mg/kg bw [36]. OCPs are highly resistant compounds; for example, DDT can remain intact in the body for 50 years. The use of DDT has also been restricted as PCBs by the Stockholm Convention since 2004, and people are now rather exposed to its metabolite DDE. Despite a nearly 50-year ban on the use of DDT in the Czech Republic (since 1974), DDE was the only one of the 67 measured pollutants that occurred in all food samples in our cohort. In addition, DDT has been shown to adversely affect not only the health of individuals who have been directly exposed but also the health status of their offspring, at least to the F2 generation [37]. DDT is known as a non-genotoxic carcinogen and endocrine disruptor. Oxidative stress seems to be a key factor in these adverse effects caused by DDT, because reactive oxygen species are generated as by-products of its metabolism [38,39,40]. In connection with pregnancy, it is also necessary to recall the fact that DDT crosses the placental barrier and may also inhibit the expression of the detoxification enzyme CYP1A1 in the placenta, thereby impairing the detoxification capacity of the placenta [41,42,43]. Insufficient protein intake is associated with a change in oxidative status [44]. Protein intake is closely related to vitamin D and B12 levels [45]. Insufficient amounts of these vitamins can lead to higher levels of oxidative stress [46,47,48,49], so this fact could be explained by the relationship between protein intake and 8-isoprostane levels.

From the other compounds analyzed in our cohort, the average daily food intake of the total PBDE congeners was 0.009 µg/kg body weight (max 1.29 µg/kg body weight. The average daily food intake of the total HBCD isomers was 0.002 µg/kg body weight (max 0.18 µg/kg body weight). The PBP food intake was 0.0015 µg/kg body weight on a single day when the PBP concentration was above LOQ. There are no weekly or daily intake limits for these substances but, based on studies, these observed doses should not have toxic effects [50]. BFRs are compounds added to products to mitigate the course of a fire. They are used in many different products, especially in electronics, textiles, and furniture. The use of BFRs was widespread in the Czech Republic thirty years ago. At present, the production of these substances in the Czech Republic is prohibited on the basis of the Stockholm Convention. However, they still occur in older products and in articles made from recycled materials. BFRs that enter the human body through skin contact, inhalation, or food consumption are relatively easy to release from various materials at elevated temperatures or in intense sunlight, so they have been found in all parts of the environment [51].

The only POP group whose food intake was different between the two locations was the PAHs. The studied localities are 300 km apart; their population and cultural customs are basically the same, so the diet and methods of food preparation also do not differ. However, air quality is different. While the average annual air BaP concentration was 0.5 ng/m^3^ in Ceske Budejovice in the year 2020, this concentration was 3 ng/m^3^ in Karvina [52,53]. Therefore, we can assume that different food PAH concentrations in both localities are due to different environmental quality, rather than dietary customs. The PAHs food concentration is controlled in various countries by limit values for either sum of PAHs or BaP. For example, the Commission Regulation (EU) 2020/1255 sets maximum levels for PAHs in food-based powders used in the preparation of beverages—BaP: 10 μg/kg; ∑BaP + BaA + BbFA + CHR: 50 μg/kg [54]. In our food samples, the average total PAH concentration was a 2.9 μg/kg food sample, the maximum measured concentration was a 47 μg/kg food sample. PAHs, widespread in the environment, are formed during the incomplete combustion of organic matter. An important route of exposure to these compounds is the ingestion of contaminated food. PAHs enter food from soil, air, and water; they are also formed directly in food during its processing at high temperatures.

The main limitation of our study is the small size of our cohort (only 54 participants). Furthermore, possible inaccuracies of the data obtained from the database “kaloricketabulky.cz”, cannot be ruled out, as it is a free online database, created and edited by volunteers. 

## 5. Conclusions

The results of our study suggest that the diet composition of Czech pregnant women should be improved. In particular, the intake of vegetable and dairy products is very low and does not reach even half of the recommended values. The vast majority of mothers do not appear to have adequate fiber intake. Education seems to be a significant socio-economic factor influencing maternal diet quality in the Czech Republic.

PCBs, OCPs, BFRs, and PAHs food intake does not exceed the recommended limits. PFAS ingestion was above the limits, but only in a limited number of subjects (four mothers (7.8%)).

Maternal protein intake and pre-pregnancy weight together explained 22% of birth weight variability in the multiple regression model. The association of both of these factors with birth weight is positive, and their contribution to birth weight variability is about the same.

Maternal DDT food intake and protein intake together explained the 40% variability of neonatal 8-isoprostane levels in the multiple regression model. DDT intake is positively associated with this marker of oxidative stress, while protein intake is negatively related. The effect of DDT intake is probably higher than the effect of protein intake.

In conclusion, pregnant women in the Czech Republic have access to good quality food and, based on the results of the analysis of the three hundred fifty-two food samples in two different locations in the Czech Republic, they are most likely not exposed to an excessive intake of POPs in their diet. However, their choice of diet composition is inappropriate and can negatively affect the child’s health. It would be appropriate to create a national program ensuring that pregnant women in the Czech Republic are informed about adequate nutrition and its importance both for their child’s health and for their own.

## Figures and Tables

**Figure 1 foods-11-03893-f001:**
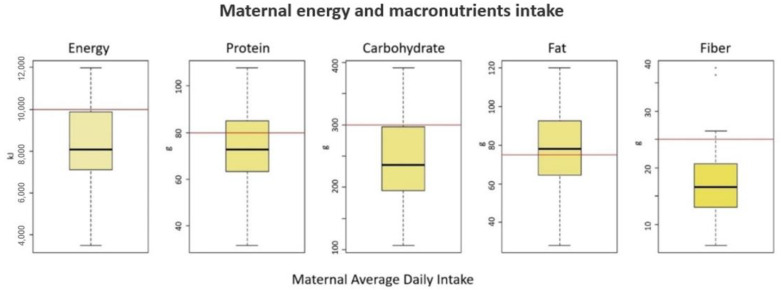
Maternal average daily intake of energy and macronutrients (protein, carbohydrate, fat, and fiber). The red horizontal lines indicate the recommended daily intake.

**Figure 2 foods-11-03893-f002:**
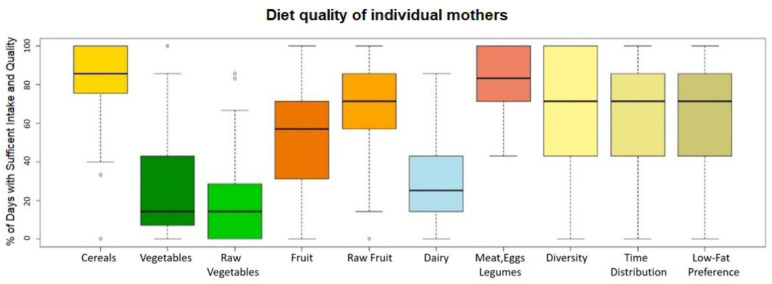
Diet quality of individual mothers.

**Figure 3 foods-11-03893-f003:**
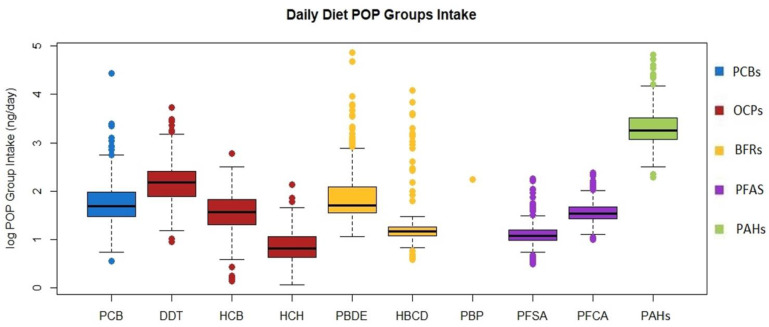
Daily intake of ∑PCB congeners, ∑DDT isomers, and their metabolites, HCB, ∑HCH isomers, ∑PBDE congeners, ∑HBCD isomers, PBP, ∑perfluorinated sulfonates, ∑perfluorinated carboxylic acids, and ∑PAHs.

**Figure 4 foods-11-03893-f004:**
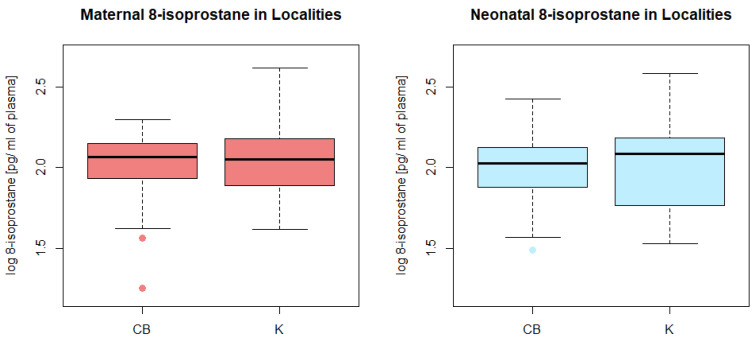
Maternal and neonatal plasma concentrations of 8-isoprostane in localities.

**Figure 5 foods-11-03893-f005:**
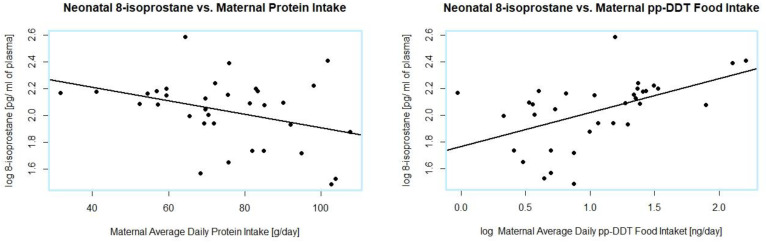
Relationships between neonatal plasma concentrations of 8-isoprostane and maternal food intake of proteins and pp-DDT.

**Figure 6 foods-11-03893-f006:**
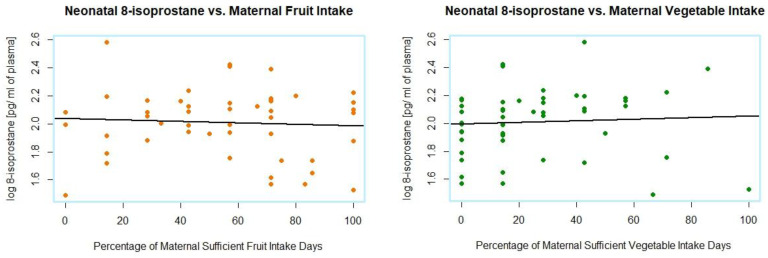
Relationships between neonatal plasma concentrations of 8-isoprostane and maternal fruit and vegetable intake.

**Table 1 foods-11-03893-t001:** Cut points of qualitative diet factors.

Diet Factor	Cut Point
Cereals	3 servings/day
Vegetables	300 g/day
Raw vegetables	200 g/day
Fruit	200 g/day
Raw fruit	100 g/day
Dairy	3 servings/day
Meat, eggs, legumes	1 serving/day
Diversity	3 different foods in 3 different food groups/day
Time distribution	5 nutritional value meals/day
Low-fat preference	predominance of low-fat foods over high-fat foods in a period of one day

**Table 2 foods-11-03893-t002:** Characteristics of the mothers and their newborns from both localities.

	Karvina			Ceske Budejovice			
	N	Mean ± SD	Median (Min.–Max.)	N	Mean ± SD	Median (Min.–Max.)	*p*
**Mothers**							
Age (years)	26	30.50 ± 5.05	30 (22–43)	28	31.50 ± 3.95	31 (25–39)	0.424
Gestation (weeks)	26	39.38 ± 1.06	40 (37–41)	28	39.32 ± 1.19	39 (37–41)	0.795
Weight (before pregnancy)	26	64.88 ± 16.66	60.50 (43–115)	28	70.82 ± 12.79	69 (48–98)	0.035
WHtR (before pregnancy)	26	0.45 ± 0.09	0.44 (0.36–0.80)	28	0.49 ± 0.08	0.47 (0.36–0.65)	0.090
BMI (before pregnancy)	26	23.57 ± 5.36	22.27 (17.21–41.23)	28	24.44 ± 4.67	22.63 (17.42–36.89)	0.350
Cotinine (ng/mg creatinine)	26	3.75 ± 7.78	1.13 (0.22–38.78)	27	1.47 ± 2.40	0.77 (0.18–12.34)	0.039
Education (Basic/ High school/University)	2/10/14	N/A	N/A	2/8/18	N/A	N/A	0.511
**Newborns**							
Birth weight (g)	26	3329 ± 411.28	3275 (2730–4170)	28	3548 ± 445.83	3540 (2650–4580)	0.067
Apgar score (value)	26	9.80 ± 0.57	10 (8–10)	28	9.79 ± 0.42	10 (9–10)	0.430
Gender (M/F)	16/10	N/A	N/A	13/15	N/A	N/A	0.401
Time interval between diet sampling and cord blood sampling (days)	26	12.77 ± 7.66	12 (1–29)	26	10.54 ±7.49	8.5 (1.5–27)	0.245

**Table 3 foods-11-03893-t003:** The relationship between diet quality and education.

	University-Educated Mothers (*n* = 33)		Lower Educated Mothers (*n* = 21)		
	Mean ± SD	Median (Min.–Max.)	Mean ± SD	Median (Min.–Max.)	*p*
Average daily intake					
Protein (g)	77.8 ± 14.6	76.9 (52.3–107.7)	68.0 ± 18.0	69.0 (31.7–103.9)	0.050
Carbohydrate (g)	258 ± 63.6	244 (147–391)	217 ± 66.6	217 (107–325)	0.036
Fiber (g)	18.7 ± 5.1	17.8 (10.6–37.6)	15.2 ± 7.5	13.1 (6.3–36.4)	0.012
% of days with sufficient intake					
Cereal (%)	86.9 ± 21.7	85.7 (0.0–100)	73.3 ± 24.0	77.4 (33.3–100)	0.024
% of days with sufficient quality					
Food intake distribution (%)	72.0 ± 28.5	71.4 (0.0–100)	54.6 ± 32.9	61.9 (0.0–100)	0.046
Low-fat food preference (%)	72.2 ± 25.7	75.0 (0.0–100)	53.3 ± 28.3	46.4 (0.0–100)	0.015
Average diet quality (max. 10 points)	5.9 ± 1.0	6.0 (4.3–7.9)	5.0 ± 1.7	5.0 (2.3–9.0)	0.038

## Data Availability

All data in our paper are available from the corresponding author upon reasonable request.

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
