# Peer review of "Maternal Diet Quality and the Health Status of Newborns"

_foods, 2022, doi:10.3390/foods11233893_

Round 1

Reviewer 1 Report

Introduction 

Lines 51-62 Authors mentioned they investigated many indicators, including the biomarker of oxidative stress, macro nutrient intake, macro nutrient intake, food intake (fruits and vegetables), and dietary POPS. What is the main question addressed by the research? What does it add to the subject area compared with other published material? 

Methods 

Lines 93-95 How to define maternal diet quality for this study? Did diet quality define based on Dietary guidelines or others? How to calculate the diet quality index? 

 Line 141- What are dietary factors? Did these include diet POPs? Did authors include all dietary factors simultaneously or create individual models?  

Results 

Line 250- Presenting Tables for associations between birth weight and diet POPS may be helpful because these are the main findings of this study. 

Figure 2 How to calculate the diet quality? What is the cut point of each food group? 

Discussion 

Limitations of this study do not seem to be described. 

Line 389- Why do the authors conclude that pregnant women in the Czech Republic have access to quality food and are not exposed to excessive intake of POPs in their diet? In Figure 2, mothers did not intake vegetables and dairy products to reach the recommendations in this study participants. Diet Pops may not be safe because the authors did not investigate child outcomes other than birth weight.  

 Author Response

We thank all reviewers for their time and valuable comments.

All text changes were proofread by a native speaker.

Comments and Suggestions for Authors

Introduction 

Lines 51-62 Authors mentioned they investigated many indicators, including the biomarker of oxidative stress, macro nutrient intake, macro nutrient intake, food intake (fruits and vegetables), and dietary POPS. What is the main question addressed by the research? What does it add to the subject area compared with other published material? 

The study is unique in that it provides more accurate information about the current quality of the diet of pregnant women in the Czech Republic based on diet records, not only from questionnaire data.

The study is also unique in that it was simultaneously obtained two groups of information about the diet of pregnant women, on the one hand from detailed diet records (quantitative and qualitative factors) and on the other hand from diet samples (POPs concentration). These two groups of information were then included simultaneously in the analysis.

 Edited text:

There were two main aims of our study. First, to examine the diet quality of pregnant women, including exposure to POPs, in the Czech Republic. Second, to analyze the relationship between maternal diet during pregnancy (including POPs intake) and the birth weight and neonatal levels of oxidative stress.

Methods 

Lines 93-95 How to define maternal diet quality for this study? Did diet quality define based on Dietary guidelines or others? How to calculate the diet quality index? 

Edited text:

The questionnaire CPT evaluates the diet quality over a 24-hour period and consists of ten ‘yes / no questions’. These questions relate to food intake from different food groups, time distribution of food intake, diet diversity and low-fat food preferences for each day (Table 1 shows the cut points for each queried qualitative factor). The total positive answers (= point gain) then indicate the overall diet quality for a given day.

Table 1.  Cut points of qualitative diet factors

Diet Factor

Cut Point

Cereals

3 servings/day

Vegetables

300 g/day

Raw vegetables

200g/day

Fruit

200g/day

Raw fruit

100g/day

Dairy

3 servings/day

Meat, eggs, legumes

1 servings/day

Diversity

3 different foods in 3 different food groups/day

Time distribution

5 nutritional value meals/day

Low-fat preference

predominance of low-fat foods over

high-fat foods in a period of one day

 For each woman, the average values of diet factors obtained from database “kaloricketabulky.cz” from all of her monitored days were calculated. The values of diet factors obtained from CPT were calculated as the percentage of days in which the woman reached or exceeded the cut point. Next, the overall diet quality was calculated for each woman as her average point gain in CPT from all the days when her diet quality was monitored. 

Line 141- What are dietary factors? Did these include diet POPs? Did authors include all dietary factors simultaneously or create individual models?

Dietary factors include quantitative and qualitative characteristics of the diet (see 2.3) as well as diet concentrations of POPs. Dietary factors were assessed simultaneously and also in individual models (see 3.4 and 3.5).

Results 

Line 250- Presenting Tables for associations between birth weight and diet POPS may be helpful because these are the main findings of this study. 

Birth weight had no associations with any diet POP in our cohort (see last sentence in 3.4.).

Figure 2 How to calculate the diet quality? What is the cut point of each food group? 

Edited text:

The questionnaire CPT evaluates the diet quality over a 24-hour period and consists of ten ‘yes / no questions’. These questions relate to food intake from different food groups, time distribution of food intake, diet diversity and low-fat food preferences for each day (Table 1 shows the cut points for each queried qualitative factor). The total positive answers (= point gain) then indicate the overall diet quality for a given day.

Table 1.  Cut points of qualitative diet factors

Diet Factor

Cut Point

Cereals

3 servings/day

Vegetables

300 g/day

Raw vegetables

200g/day

Fruit

200g/day

Raw fruit

100g/day

Dairy

3 servings/day

Meat, eggs, legumes

1 servings/day

Diversity

3 different foods in 3 different food groups/day

Time distribution

5 nutritional value meals/day

Low-fat preference

predominance of low-fat foods over

high-fat foods in a period of one day

For each woman, the average values of diet factors obtained from database “kaloricketabulky.cz” from all of her monitored days were calculated. The values of diet factors obtained from CPT were calculated as the percentage of days in which the woman reached or exceeded the cut point. Next, the overall diet quality was calculated for each woman as her average point gain in CPT from all the days when her diet quality was monitored.  

Discussion 

Limitations of this study do not seem to be described. 

Added text:

The main limitation of our study is the small size of our cohort (only 54 participants). Furthermore, possible inaccuracies of the data obtained from the database "kaloricketabulky.cz", cannot be ruled out, as it is a free online database, created and edited by volunteers.

Line 389- Why do the authors conclude that pregnant women in the Czech Republic have access to quality food and are not exposed to excessive intake of POPs in their diet? In Figure 2, mothers did not intake vegetables and dairy products to reach the recommendations in this study participants. Diet Pops may not be safe because the authors did not investigate child outcomes other than birth weight.  

Diet POPs did not exceed legal limits. Relationships between diet POPs and newborn oxidative damage was but also analyzed. DDT was the only POPs that showed a statistical association with newborn oxidative damage. However, its concentrations were several orders of magnitude lower than the limit (see the first sentence of the seventh paragraph of the Discussion).

Added text:

In the Czech Republic, good quality food is available all year round in a very dense retail network. The Czech Republic is a developed welfare state with many social benefits. The vast majority of the population have the financial means to buy good quality food. All study participants were asked about their financial situation. No one reported financial problems that would force them to limit their living needs. Thus there are no external obstacles preventing women from accessing good quality food here.

Reviewer 2 Report

The work reinforces the importance of education about the effects of maternal nutrition on the development of their children, with appropriate references. They also investigated the effect of pollution on the diet, which is a very important point and deserves further investigation, but found no significant differences. I believe that this was due to the fact that the sample size is too small to divide it into so many study groups, taking into account the dispersion of the variable.

Compared with other published material, it does not add new concepts but reinforces the importance of educating about the importance of a healthy diet during pregnancy, especially in countries where food availability is great, as well as the need for local repopulation.

I would recommend increasing the size of the sample to be able to show the effects of the contaminants that, in my opinion, remained hidden.

The conclusions related to the quality of the maternal diet, in relation to its nutritional composition, are valued correctly.
“…pregnant women in the Czech Republic have access to quality food and are not exposed to excessive intake of POPs in their diet.” They should not express it in such a forceful way having evaluated only 54 women.

There are also problems with the form of some figures and tables, e.g.:
Table 1. Incomplete title
Fig.2. If each box represents the consumption of all the mothers, the epigraph is incorrect. The title must be removed and in any case complete the epigraph with that information.
Table 2. The title could be clearer. They do not mention how many mothers are in each group.
Fig.3. The references to the right are not necessary because the abscissa axis is already indicated. The title must be removed and in any case complete the epigraph with that information.
Fig. 4. “Maternal and neonatal plasma concentrations of 8-isoprostane in localities”. Which box is from each locality???
The titles above each box should be replaced by the name of the locality.

Author Response

We thank all reviewers for their time and valuable comments.

All text changes were proofread by a native speaker.

Comments and Suggestions for Authors

The work reinforces the importance of education about the effects of maternal nutrition on the development of their children, with appropriate references. They also investigated the effect of pollution on the diet, which is a very important point and deserves further investigation, but found no significant differences. I believe that this was due to the fact that the sample size is too small to divide it into so many study groups, taking into account the dispersion of the variable.

Compared with other published material, it does not add new concepts but reinforces the importance of educating about the importance of a healthy diet during pregnancy, especially in countries where food availability is great, as well as the need for local repopulation.

I would recommend increasing the size of the sample to be able to show the effects of the contaminants that, in my opinion, remained hidden.

The conclusions related to the quality of the maternal diet, in relation to its nutritional composition, are valued correctly.
“…pregnant women in the Czech Republic have access to quality food and are not exposed to excessive intake of POPs in their diet.” They should not express it in such a forceful way having evaluated only 54 women.

Added text:

In the Czech Republic, good quality food is available all year round in a very dense retail network. The Czech Republic is a developed welfare state with many social benefits. The vast majority of the population have the financial means to buy good quality food. All study participants were asked about their financial situation. No one reported financial problems that would force them to limit their living needs. Thus there are no external obstacles preventing women from accessing good quality food here.

Edited text:

In conclusion, pregnant women in the Czech Republic have access to good quality food and, based on the results of the analysis of the three hundred and fifty-two food samples in two different locations in the Czech Republic, they are most likely not exposed to excessive intake of POPs in their diet. However, their choice of diet composition seems to be inappropriate and can negatively affect the child's health.

There are also problems with the form of some figures and tables, e.g.:
Table 1. Incomplete title

Edited title of Table 1:

Characteristics of the mothers and their newborns from both localities

Edited figure:

Fig.2. If each box represents the consumption of all the mothers, the epigraph is incorrect. The title must be removed and in any case complete the epigraph with that information.

Edited figure:

Table 2. The title could be clearer. They do not mention how many mothers are in each group.

Edited table:

Table 2 (now Table 3). The relationship between diet quality and education

University-educated mothers (N=33)

Lower educated mothers (N=21)

Mean ± SD

Median (Min. – Max.)

Mean ± SD

Median (Min. - Max.)

p

Average daily intake

Protein (g)

77.8 ± 14.6

76.9 (52.3 107.7)

68.0 ± 18.0

69.0 (31.7 103.9)

0.050

Carbohydrate (g)

258 ± 63.6

244 (147 391)

217 ± 66.6

217 (107 325)

0.036

Fiber (g)

18.7 ± 5.1

17.8 (10.6 37.6)

15.2 ± 7.5

13.1 (6.3 36.4)

0.012

% of days with sufficient intake

Cereal (%)

86.9 ± 21.7

85.7 (0.0 100)

73.3 ± 24.0

77.4 (33.3 100)

0.024

% of days with sufficient quality

Food intake distribution (%)

72.0 ± 28.5

71.4 (0.0 –100)

54.6 ± 32.9

61.9 (0.0 – 100)

0.046

Low-fat food preference (%)

72.2 ± 25.7

75.0 (0.0 –100)

53.3 ± 28.3

46.4 (0.0 – 100)

0.015

Average diet quality (max. 10 points)

5.9 ± 1.0

6.0 (4.3 7.9)

5.0 ± 1.7

5.0 (2.3 9.0)

0.038

Fig.3. The references to the right are not necessary because the abscissa axis is already indicated. The title must be removed and in any case complete the epigraph with that information.

On the axis of the abscissa there are 10 groups of substances grouped according to their chemical structure. These groups are then further grouped into 5 groups normally distinguished in the environmental sector. These five groups are marked on the link to the right. It is therefore a double grouping – chemical (axis of the abscissa) and environmental (links on the right).

Fig. 4. “Maternal and neonatal plasma concentrations of 8-isoprostane in localities”. Which box is from each locality???
The titles above each box should be replaced by the name of the locality.

The locality designation is below each box.

Reviewer 3 Report

Thank you for the opportunity to review this manuscript. The main focus of the manuscript was to examine the dietary quality of pregnant women, including exposure to pollutants, and to test the association of these factors with birth weight and oxidative stress.

The comments are listed below:

Specifically:

Section 2.2 – indicated that women provided food records for 7 days in the period of two weeks prior to their birth due date. Clarification on when these records occurred would be beneficial. For example, was it for 7 consecutive days so that every day of the week was represented? If not, which days were represented?

Can a more detailed description be provided for the process of using “ kaloricketabulky.cz” database? For example, what, if any, assumptions/substitutions were made during the analysis process? If the food was not in the database, how did the research team approach this issue?

Line 87- indicates that maternal weight and waist circumference was obtained in personal interviews with the mothers during their stay in the maternity hospitals. Were these pre-pregnancy values which were reported in lines 158-163 of the results?

Figure 1 and 2 would benefit from a more in-depth figure description/legend. For figure 1 – it is not clear what the red horizontal lines are indicating. Figure 2 – more description around the axis would be beneficial (for example is the y axis referring to % of days with sufficient intake and quality?).

Time distribution measure – it is unclear how this was determined. This clarification could be included in the methods section, or in the results when discussing figure 2.

Birth weight may also be impacted by maternal weight gain during pregnancy. Was weight gain during pregnancy measured? If not, a discussion about the influence of maternal weight gain during pregnancy and its effects on birth outcome warrants a mention in the discussion section.

Line 342 – a “)” is missing before the end of the sentence.

Author Response

We thank all reviewers for their time and valuable comments.

All text changes were proofread by a native speaker.

Comments and Suggestions for Authors

Thank you for the opportunity to review this manuscript. The main focus of the manuscript was to examine the dietary quality of pregnant women, including exposure to pollutants, and to test the association of these factors with birth weight and oxidative stress.

The comments are listed below:

Specifically:

Section 2.2 – indicated that women provided food records for 7 days in the period of two weeks prior to their birth due date. Clarification on when these records occurred would be beneficial. For example, was it for 7 consecutive days so that every day of the week was represented? If not, which days were represented?

Edited text:

Pregnant women recorded food and drink consumption on pre-prepared forms for seven consecutive days…

 Can a more detailed description be provided for the process of using “ kaloricketabulky.cz” database? For example, what, if any, assumptions/substitutions were made during the analysis process? If the food was not in the database, how did the research team approach this issue?

Added text:

If the some food was not in the database, it was replaced by the most similar equivalent.

Line 87- indicates that maternal weight and waist circumference was obtained in personal interviews with the mothers during their stay in the maternity hospitals. Were these pre-pregnancy values which were reported in lines 158-163 of the results?

Yes, it was the pre-pregnancy values that are shown in lines 158-163 of the results.

Edited text:

Socioeconomic data and data on maternal height, pre-pregnancy weight, weight before delivery and pre-pregnancy waist circumference were obtained in personal interviews with the mothers during their stay in the maternity hospitals. 

Figure 1 and 2 would benefit from a more in-depth figure description/legend. For figure 1 – it is not clear what the red horizontal lines are indicating. Figure 2 – more description around the axis would be beneficial (for example is the y axis referring to % of days with sufficient intake and quality?).

Edited Figure 1 description:

 Figure 1. Maternal average daily intake of energy and macronutrients (protein, carbohydrate, fat, fiber). The red horizontal lines indicate the recommended daily intake.

Edited Figure 2:

Time distribution measure – it is unclear how this was determined. This clarification could be included in the methods section, or in the results when discussing figure 2.

Added new table 1:

 Cut points of qualitative diet factors:

Diet Factor

Cut point

Cereals

3 servings/day

Vegetables

300 g/day

Raw vegetables

200g/day

Fruit

200g/day

Raw fruit

100g/day

Dairy

3 servings/day

Meat, eggs, legumes

1 servings/day

Diversity

3 different foods in 3 different food groups/day

Time distribution

5  nutritional value meals/day

Low-fat preference

predominance of low-fat foods over high-fat foods in a period of one day

Birth weight may also be impacted by maternal weight gain during pregnancy. Was weight gain during pregnancy measured? If not, a discussion about the influence of maternal weight gain during pregnancy and its effects on birth outcome warrants a mention in the discussion section.

 Yes, maternal weight gain during pregnancy was measured. It did not show a statistically significant relationship with birth weight (p = 0.453).

Added text:

Birth weight also had no relationship with maternal weight gain.

 Line 342 – a “)” is missing before the end of the sentence.

Edited text:

… variability was explained by pp-DDT and protein intake (p < 0.001).